# Mesenchymal Stromal Cells from Perinatal Tissues as an Alternative for Ex Vivo Expansion of Hematopoietic Progenitor and Stem Cells from Umbilical Cord Blood

**DOI:** 10.3390/ijms242115544

**Published:** 2023-10-24

**Authors:** Ximena Bonilla, Ana Milena Lara, Manuela Llano-León, David A. López-González, David G. Hernández-Mejía, Rosa Helena Bustos, Bernardo Camacho-Rodríguez, Ana-María Perdomo-Arciniegas

**Affiliations:** 1Pharmaceutical Biotechnology Unit, Instituto Distrital de Ciencia, Biotecnología e Innovación en Salud, Bogotá 111611, Colombia; amlara@idcbis.org.co (A.M.L.); dlopez@idcbis.org.co (D.A.L.-G.); bacamacho@idcbis.org.co (B.C.-R.); 2Advanced Therapies Unit, Instituto Distrital de Ciencia, Biotecnología e Innovación en Salud, Bogotá 111611, Colombia; mllano@idcbis.org.co (M.L.-L.); dghernandez@idcbis.org.co (D.G.H.-M.); 3Therapeutic Evidence Group, Clinical Pharmacology, Universidad de La Sabana and Clínica Universidad de La Sabana, Chía 140013, Colombia; rosa.bustos@unisabana.edu.co; 4Cord Blood Bank, Instituto Distrital de Ciencia, Biotecnología e Innovación en Salud, Bogotá 111611, Colombia; amperdomo@idcbis.org.co

**Keywords:** hematopoietic stem and progenitor cells ex vivo expansion, mesenchymal stromal cells, perinatal tissues, umbilical cord blood, hematopoietic stem and progenitor cells transplantation

## Abstract

Umbilical cord blood (UCB) serves as a source of hematopoietic stem and progenitor cells (HSPCs) utilized in the regeneration of hematopoietic and immune systems, forming a crucial part of the treatment for various benign and malignant hematological diseases. UCB has been utilized as an alternative HSPC source to bone marrow (BM). Although the use of UCB has extended transplantation access to many individuals, it still encounters significant challenges in selecting a histocompatible UCB unit with an adequate cell dose for a substantial proportion of adults with malignant hematological diseases. Consequently, recent research has focused on developing ex vivo expansion strategies for UCB HSPCs. Our results demonstrate that co-cultures with the investigated mesenchymal stromal cells (MSCs) enable a 10- to 15-fold increase in the cellular dose of UCB HSPCs while partially regulating the proliferation capacity when compared to HSPCs expanded with early acting cytokines. Furthermore, the secretory profile of UCB-derived MSCs closely resembles that of BM-derived MSCs. Moreover, both co-cultures exhibit alterations in cytokine secretion, which could potentially impact HSPC proliferation during the expansion process. This study underscores the fact that UCB-derived MSCs possess a remarkably similar supportive capacity to BM-derived MSCs, implying their potential use as feeder layers in the ex vivo expansion process of HSPCs.

## 1. Introduction

Umbilical cord blood (UCB) has been employed as an alternative source of hematopoietic stem and progenitor cells (HSPCs) and is utilized for transplantation in compatible patients lacking a family donor, achieving hematological reconstitution through various conditioning regimens. HSPCs have found extensive application in the treatment of lymphomas, leukemias, and other chronic diseases of the hematopoietic system [1,2,3]. Successful HSPC reconstitution involves several steps: HSPCs migrate to the microenvironment or “niche” of the bone marrow (BM), guided by diverse biological chemotactic mediators such as stromal cell-derived factor 1 (SDF-1), prostaglandin E2 (PGE2), and others. Subsequently, HSPCs engraft, expand, and occupy the medullary “niche”. Finally, HSPCs differentiate in response to various stimuli, thus reconstituting the hematopoietic system comprising neutrophils, red blood cells, platelets, lymphocytes, and more [2].

HSPCs can be derived from three primary sources: BM, mobilized peripheral blood, and UCB. UCB presents advantages in terms of recipient–donor histocompatibility restrictions, coupled with a lower incidence of graft-versus-host disease, attributed to the immunological immaturity of the donor [4,5]. Furthermore, UCB collection occurs non-invasively at birth, posing no harm to the newborn. Subsequent freezing allows for storage and preservation, ensuring its availability over time through blood banks for patients [6,7,8].

Despite significantly improved transplantation access for numerous individuals who otherwise would not benefit from this treatment, the utilization of UCB still confronts substantial challenges. Selecting a histocompatible UCB unit with an adequate cell dose remains a hurdle for a considerable proportion of adults with malignant hematological diseases [9]. The low cell dose in UCB units, in terms of both total nucleated cells (TNCs), a determinant based on the patient’s weight for transplantation, and CD34+ cells, results in prolonged engraftment times (25 days), as compared to other transplant types like BM (17 days) or mobilized peripheral blood (14 days) [1,2]. These extended engraftment periods can lead to short- and long-term infectious complications [8,10,11]. As a consequence, UCB’s application is primarily confined to pediatric patients [11], limiting its use in adults unless methods are employed to compensate for the dose of infused cells [2,8,12].

Several strategies have been proposed to address the cell dose limitation, one of which involves the transplantation of two UCB units. However, research has revealed that only one of the transplanted units is capable of engrafting and regenerating the hematopoietic system [1,8]. Another approach is the ex vivo manipulation of UCB cells before transplantation [2,3,13]. This manipulation may include genetic modifications to stimulate signaling pathways (c-KIT, NOTCH, PGE2) in one of the units. Alternate strategies involve expanding HSPCs using cytokines, copper chelation, or co-culturing with mesenchymal stromal cells (MSCs) [1,14].

The ex vivo expansion of cord blood cells has been explored from various perspectives based on the physiological mechanisms within the medullary niche that balance HSPC maintenance and regeneration. For example, treatments with cytokines present in the niche, blocking different adhesion molecules, and the inhibition of DNA methylation have been investigated [10,15]. These strategies promote an increase in the number of cells available for transplantation. However, they lead to the expansion of HSPCs with a certain degree of differentiation rather than the desired primitive state [10,14]. This ultimately reduces the capacity for engraftment and immune reconstitution, which are critical for the recovery of patients undergoing transplantation [2], among other factors.

The concept of the “medullary niche” has been studied and was initially defined as an anatomical and functional microenvironment within the bone marrow occupied by HSPCs, where the surroundings enable them to remain undifferentiated [16]. Subsequent studies have revealed that within the medullary niche, progenitors interact with several cell types (endothelial cells, immune cells, peripheral nervous system neurons, osteoblasts, osteoclasts, mesenchymal stromal cells, and hematopoietic cells), components of the extracellular matrix, and soluble factors that regulate their activities and determine the timing of processes such as differentiation, quiescence, or proliferation, in response to physiological stimuli and systemic needs [3,8]. Based on these interactions, two ex vivo expansion techniques for HSPCs have been explored. Firstly, HSPCs are isolated from fresh or frozen hematopoietic tissue and incubated with a combination of early acting cytokines involved in HSPC homeostasis and proliferation, including stem cell factor (SCF), FMS-like tyrosine kinase 3 ligand (FLT3-L), and thrombopoietin (TPO). These cytokines, collectively referred to as early acting cytokines, synergistically stimulate HSPC proliferation ex vivo, mimicking the conditions of the medullary microenvironment [1,14,17]. The second expansion technique involves co-culturing with MSCs. In the niche, MSCs interact with HSPCs either through the secretion of soluble factors into the microenvironment or through direct contact between the cell populations via surface proteins. This interaction supports the maintenance of the stem properties of HSPCs [18]. Furthermore, it has been demonstrated that MSCs can ex vivo maintain the quiescent state of BM-isolated HSPCs through these interactions [10,19].

MSCs are multipotent cells that give rise to the components of connective tissue in adult organisms, including bones, cartilage, tendons, ligaments, muscles, and bone marrow. Consequently, they possess adipogenic, chondrogenic, and osteogenic differentiation potential and are characterized by their fibroblastic morphology [20,21]. Their identification relies on their in vitro plastic-adherence property in culture and their immunophenotypic characterization via flow cytometry based on the criteria established by the International Society for Cellular Therapy (ISCT). This involves the expression of specific markers, CD105, CD90, CD73, CD44, and HLA-I, while lacking markers such as CD19, CD45, CD34, CD14, and HLA-II [19,22]. MSCs can be harvested from various tissues in the adult human body, including BM, peripheral blood, dental pulp, adipose tissue, menstrual blood, endometrium, and breast milk. Similarly, perinatal tissues like amniotic fluid, membranes, placenta, and umbilical cord (including Wharton’s jelly and UCB) can serve as sources of MSCs. Notably, MSCs derived from perinatal tissues exhibit a faster proliferation rate compared to those isolated from adult human tissues [19,23]. It has been demonstrated that MSCs derived from BM, UCB, and Wharton’s jelly secrete various cytokines and growth factors of significant importance, including granulocyte colony-stimulating factor (G-CSF), granulocyte-macrophage colony-stimulating factor (GM-CSF), and interleukins (IL-6 and IL-8). These molecules appear to play roles in the proliferation, survival, and differentiation of HSPCs within the medullary niche [24,25].

In this context, several studies have proposed a co-culture technique involving BM-derived MSCs supplemented with early acting cytokines for the ex vivo expansion of UCB HSPCs. This approach more closely mimics the conditions of the niche, enabling the increase of the HSPC cell dose by 8 to 37 times while maintaining them in an undifferentiated state [13,14]. However, its clinical application remains limited, necessitating further optimization and scalability of this technique.

This study aims to develop an ex vivo expansion protocol for HSPCs isolated from UCB, co-cultured with MSCs from perinatal tissues (Wharton’s jelly and UCB), and compare the use of BM-MSC that has proven to be useful for progenitor expansion. This approach is driven by the relatively unexplored potential of these tissues as sources of MSCs, which can serve as support for expansion through the release of soluble factors into the medium or direct cell-to-cell contact. This knowledge contributes to the exploration of alternative MSC support sources and, more broadly, advances the development of standardized expansion techniques on a larger scale.

## 2. Results

### 2.1. Isolation and Characterization of Mesenchymal Stromal Cells from Bone Marrow and Perinatal Tissues

To ensure the purity and proper functionality of MSCs isolated from different tissues, tests defined by the International Society for Cellular Therapy (ISCT) were performed, which related to phenotypic and immunophenotypic characterization. 

In this study, MSCs isolated from BM, UCB, and WJ exhibited the properties of plastic adherence and a fibroblastoid morphology (Appendix A). Additionally, these cells displayed high expression levels for the markers CD73, CD90, CD105, and CD44 and low expression for the hematopoietic and endothelial markers CD45 and CD54, respectively (Appendix A). It is important to note that the UCB source of MSC variably expressed the immune marker HLA-DR (≤24%). While MSCs are typically expected to be HLA-DR-negative, it has been observed that under certain stimuli, these cells can express HLA-DR surface molecules due to their immunomodulatory properties. Therefore, in line with the guidelines of the ISCT, these cells can still be classified as MSCs [21,22]. Recent studies have demonstrated that the dynamic expression of HLA-DR in clinical-grade MSC cultures does not affect the in vitro phenotype or functionality of these cells, thereby supporting their continued classification as MSCs [26].

### 2.2. Definition of Culture Conditions for Hematopoietic Stem and Progenitor Cell Expansion

To establish consistent culture conditions across all experiments, various variables were evaluated, including culture time, the presence or absence of early-action cytokines in the culture, MSC confluence percentage, the impact of gamma irradiation on the MSC’s support capacity, and their secretome. Appendix A illustrates that the addition of early-action cytokines has a synergistic effect on HSPC proliferation. Furthermore, it can be observed that gamma irradiation does not induce statistically significant changes in either the proliferation capacity of HSPCs or the secretome of MSCs. Nonetheless, it is considered an important preconditioning technique to maintain MSC confluence during the co-culture process. Regarding the co-culture duration, the proliferation capacity was evaluated at 3, 7, and 9 days. In this context, it was established that seven days of culture is adequate for assessing the proliferation capacity of these cells without inducing excessive cell growth or complete nutrient depletion within the co-culture. The definition of these variables is primarily rooted in the in vitro emulation of the hematopoietic niche.

### 2.3. Evaluation of the Supportive Capacity of Perinatal Mesenchymal Stromal Cells in Hematopoietic Stem- and Progenitor-Cell Proliferation

Previous studies have observed that the co-culture of HSPCs with BM-derived MSCs, serving as an ex vivo expansion model, not only enhances the proliferation of HSPCs but also contributes to the regulation of their functional properties. This outcome is closely linked to the supportive effect exerted by MSCs on HSPCs within the bone marrow niche.

In this study, the proliferation capacity of HSPCs co-cultured with BM and perinatal-tissue-derived MSCs was examined. Proliferation was quantified through manual counting and the calculation of proliferation percentage, as well as by determining the Mean Fluorescence Intensity of the CFSE probe using flow cytometry. A comparison of the proliferation rate of HSPCs co-cultured with MSCs derived from BM and umbilical cord blood (UCB) to those cultured with early-action cytokines (HSPC+Cytokines) reveals a general increase in the proliferation of HSPCs co-cultured with BM- and UCB-derived MSCs, underscoring the regulatory role of MSCs on HSPCs. However, this proliferation capacity is less pronounced when co-cultured with WJ-derived MSCs. Figure 1A,B (paired assays) depict the proliferation capacity of HSPCs from the same donor once co-cultured in the presence of MSCs. Notably, proliferation increases partially with BM and UCB-derived MSCs, while there is a reduction in proliferation for HSPCs co-cultured with WJ-derived MSCs. This trend is consistent when comparing multiple donor combinations of HSPCs and MSCs, with BM- and UCB-derived MSCs maintaining HSPCs proliferation capacity, whereas WJ-derived MSCs exhibit a diminished capacity. In terms of proliferation determination through CFSE labeling, the MFI of HSPCs co-cultured with BM- and UCB-derived MSCs supports their proliferation, whereas WJ-derived MSCs limit HSPCs proliferation during co-culture.

### 2.4. Evaluation of the Migratory Profile of Hematopoietic Stem and Progenitor Cells in Co-Culture with Mesenchymal Stromal Cells from Perinatal Tissues

Assessing the responsiveness of HSPCs to chemotactic stimuli inherent to the bone marrow niche (SDF-1 and VCAM-1) is a crucial parameter for determining the homing and engraftment capacity of these cells, which serves as a critical criterion for their primitiveness and the success of transplanted expanded HSPCs.

Figure 2 illustrates the migratory profile of both freshly isolated HSPCs and those that have undergone cellular expansion with MSCs, utilizing an in vitro migration model. Upon comparison with freshly isolated cells, no statistically significant changes are observed in the migratory pattern of cells cultured with MSCs from different tissues. Similarly, when comparing perinatal-tissue-derived MSCs to the reference pattern of BM, no significant differences in migration capacity relative to bone marrow are observed. However, a trend toward a reduction in migration capacity is noted for all three co-culture types. As with the other functional assays, the data exhibit a high degree of dispersion, likely stemming from the inherent interindividual variability among the HSPCs donor samples.

### 2.5. Determination of the Expression of Primitive and Lineage-Committed Markers in Hematopoietic Stem Progenitor Cells Cultured with Mesenchymal Stromal Cells from Perinatal Tissues

One of the primary challenges in several ex vivo expansion models for HSPCs is the occurrence of differentiation events during the process, leading to a loss in the hematopoietic reconstitution capacity of patients undergoing transplantation. To assess the preservation of HSPCs’ stemness following the expansion process with mesenchymal stromal cells from perinatal tissues, changes in the expression of the primitive marker (CD34) and the hematopoietic differentiation marker (CD38), as well as markers for lineage-committed progenitors of the myeloid (CD123) and lymphoid (CD7, and CD3) lineages, were evaluated at both day zero and seven days after co-culture.

Figure 3 illustrates the variations in the CD34 and CD38 markers, revealing that no statistically significant differences were observed for the CD34 marker. However, under MSC co-culture conditions, a slightly greater preservation of this marker is observed compared to cells in monoculture. Conversely, variations in the expression of the CD38 marker, which distinguishes lineage-committed progenitors from more primitive ones (CD34), were observed after seven days of co-culture across all tested conditions compared to day 0. This distinction was more pronounced in the case of cells in monoculture as well as those co-cultured with UCB-derived MSCs.

Evaluation of the expression of lineage-committed markers (Appendix A) revealed that there were no significant variations in the expression of markers corresponding to the myeloid lineage CD123, nor in the expression of markers for cells of the lymphoid lineage (CD3) and T-cell type (CD7). Despite the absence of significant immunophenotypic changes during the expansion culture, there is an apparent tendency towards increased expression of CD3 and CD7. For instance, CD123 expression was maintained when comparing all treatments. However, similar to the other functional assays presented earlier, data variability is evident, primarily attributed to the inherent variability among donors from the two studied populations.

### 2.6. Evaluation of the Clonogenic Potential of Hematopoietic Stem Progenitor Cells in Co-Culture with Mesenchymal Stromal Cells from Perinatal Tissues

To determine the functionality of HSPCs after undergoing the expansion process and assess their differentiation potential, the clonogenic capacity of HSPCs was evaluated (Figure 4). In this case, it was found that when HSPCs were cultured after 7 days in the colony-inducing medium, these cells were capable of proliferating and differentiating into different specific lineages (GMM-FCU, GM-CFU, G-CFU, and M-CFU). Furthermore, it is observed that despite not reaching statistical significance, MSCs from different tissues, primarily BM-MSCs, are able to maintain this clonogenic capacity when compared to HSPCs cultured solely with early-action cytokines, doubling the number of colonies. This result may suggest that MSCs could be contributing to the maintenance of HSPCs functionality.

### 2.7. Evaluation of the Secretory Profile of Hematopoietic Stem Cells in Co-Culture with Mesenchymal Stromal Cells from Perinatal Tissues

It has been previously described that the interaction between BM-derived MSCs and HSPCs in ex vivo models favors the secretion of molecules primarily related to proinflammatory mechanisms and cellular differentiation. Depending on the context, this interaction contributes to maintaining the stem properties of HSPCs, partially simulating the hematopoietic niche model. This group of molecules exerts a significant influence on the processes of proliferation and differentiation of HSPCs in ex vivo cell-expansion models.

In this study, the secretion of specific molecules involved in the interaction between HSPCs and MSCs in the bone marrow niche was quantified using the multiparametric suspension luminometry assay (Luminex, Austin, TX, USA) (Figure 5). The objective was to associate changes in the secretory profile of these cells with the functional properties evaluated during the process of cell expansion. The molecules from the MSC secretome (MCP-1, IL-6, IL-8, CCL5, VEGF, FGF-b) and the molecules secreted by HSPCs (GM-CSF and G-CSF) were assessed. These molecules were evaluated in cells treated solely with early acting cytokines (HSPC+cytokines), as well as in HSPCs co-cultured with BM-derived MSCs and perinatal-tissue-derived MSCs. The aim was to identify the molecules that might be involved in the regulation of the proliferation, migration, and differentiation of HSPCs.

Initially, the secretion of this group of cytokines was evaluated in MSCs in monoculture after irradiation, aiming to prevent MSC overgrowth during the cell-expansion process and to determine if this irradiation effect could modify the secretory profile of MSCs. Appendix A shows that this process had no significant effects compared to non-irradiated MSCs under the defined conditions of the expansion process. Thus, in this case, the irradiation process did not directly affect the functionality of MSCs or the proliferation of HSPCs during ex vivo expansion.

Figure 5A–C illustrates the secretory profiles of both MSCs in monoculture and MSCs in co-culture with HSPCs. Co-culturing with BM-derived MSCs resulted in a 2–8 times increase in all evaluated cytokines compared to monoculture cells. Notable differences were observed in the secretory profile of co-cultures with WJ-derived MSCs compared to BM, with specific increases in cytokine concentrations such as FGF, G-CSF, GM-CSF, IL-6, IL-8, and VEGF. Similarly, cultures with UCB-derived MSCs exhibited an elevated expression of cytokines such as CCL-2, EGF, FGF, G-CSF, GM-CSF, IL-6, IL-8, and VEGF. Among the three sources of MSCs, BM-derived cells exhibited the highest cytokine secretion, followed by UCB and WJ. This pattern could be directly related to the capacity to support and promote HSPCs proliferation.

In contrast, studying the secretory profile of HSPCs cultured solely with early acting cytokines revealed negligible secretion. Importantly, comparing MSCs in monoculture to cells in co-culture underscores the significant impact of direct interaction between these two cellular populations on promoting HSPC proliferation.

To comprehend the relationship between MSC sources and their secretory profile, a clustering analysis was conducted using the pheatmap tool (Figure 5D,E). Initially, the analysis focused on the relationship between cells in monoculture, both MSCs and HSPCs, and cells in co-culture, regardless of the source. In this context, data grouping occurred as follows: cells in monoculture, both MSCs and HSPCs, exhibited insignificant increases in evaluated cytokines. However, when these cells came into contact, there was a significant rise in cytokine secretion associated with MSCs.

When specifically evaluating the secretory profiles according to the donor of MSCs from different tissues and expanded HSPCs, it was observed that the secretory profiles of BM and UCB MSCs clustered similarly, secreting cytokines such as VEGF, IL-6, IL-8, CCL2, and EGF. In contrast, WJ MSCs clustered separately, particularly in terms of the secretion of cytokines like GM-CSF, G-CSF, and FGF. This once again highlights the distinction between MSC sources that has been observed in other assessed cellular properties.

Finally, the clustering strategy was applied to groups of non-irradiated and irradiated MSCs (Appendix A). In this case, the algorithm could not identify a discernible pattern to differentiate the samples based on their secretory profile. Thus, it is concluded that the irradiation protocol used does not have an identifiable effect on the secretion profile of the samples.

### 2.8. Principal Component Analysis for Assessing the Impact on Functionality Variables of HSPCs during Co-Culture with Perinatal-Tissue-Derived MSCs

Principal component analysis was performed using all the data obtained during the study to determine the impact of each variable on the co-culture process of MSCs with perinatal tissues and BM, as well as to identify clustering patterns. In this analysis, a significant contribution of the secretory profile to the first principal component was observed, especially in the secretion of G-CSF, EGF, and FGFb (Figure 6A). On the other hand, variables that made a greater contribution to the second principal component were the secretion of CCL2, the migration percentage, and the expression of CD34 and CD38 markers on day 0. Notably, the proliferation rate and secretion of IL-8 contribute significantly to both principal components captured by the analysis.

The PCA of individuals clearly identified two principal components that divided the data into two main groups based on the evaluated treatments (Figure 6B): on one side, the results associated with the co-culture of HSPCs with WJ-MSC were grouped (red cluster), and on the other side, the results associated with the treatments of HSPCs +cytokines, BM-MSC, and UCB-MSC were grouped (blue cluster). This very tendency was independently observed in the proliferation, differentiation, and secretory profile assays of the HSPCs. This analysis could indicate that the functionality of HSPCs cultured with UCB is similar to that of co-culture with BM, serving as a reference pattern in ex vivo expansion techniques.

## 3. Discussion

Previous studies have demonstrated that the ex vivo expansion of UCB HSPCs through co-culture with BM MSCs is an important strategy for increasing the number of CD34+ cells available for hematopoietic progenitor transplantation [7]. However, the availability of BM MSCs is limited due to the complexity of obtaining the sample and the nature of the tissue. For this reason, this study evaluated the supportive capacity of MSCs from perinatal tissues, which are characterized by higher availability, ease of isolation, and expansion in culture (WJ and UCB). In general, a comparison was made between the supportive capacity of MSCs from perinatal tissues and the standard expansion strategy involving co-culture with BM, aiming to determine if these cells exhibit similar or improved support capability while primarily maintaining their proliferation, migration, and differentiation properties. This was performed to pose novel cell-expansion alternatives with potential clinical use.

Throughout the study, a high degree of data dispersion was observed corresponding to the functionality of the ex vivo expansion system. This dispersion can be attributed to several factors. Firstly, there is the inherent variability in both the donors of the MSCs and the donors of the HSPCs, which has been consistently observed in other studies [7]. Secondly, there are the culture conditions of the MSCs, specifically the number of passages of the MSCs, which has also been reported to induce changes in the secretory profile of the cells [21], and similarly, changes in the functionality of the HSPCs. However, despite the variability in the aforementioned data, we could identify tendencies that corresponded to clear variations in the functionality of the HSPCs after the expansion process. Additionally, other tendencies also became evident when grouping the data according to the evaluated secretory profile (Figure 5D,E) and comprehensively considering all the variables studied (Figure 6), which effectively discriminates the data according to the source of the MSCs.

Initially, we defined culture conditions for the development of the ex vivo expansion strategy. Firstly, we evaluated the effect of gamma irradiation on MSCs, as this process has been described to inhibit MSC proliferation while preserving cell viability and functionality [27], thereby enhancing the supportive capacity of MSCs at low radiation doses and increasing the proliferation capacity of HSPCs in an ex vivo co-culture expansion system [28]. Here, we studied whether the irradiation of MSCs prior to co-culture had any effect on both the expansion process of HSPCs and the functionality of MSCs. It was found that irradiation did not have a significant effect on the secretory profile of HSPCs, indicating that they retain similar abilities in inducing HSPCs proliferation (Appendix A). On the other hand, the role of early acting cytokines in the culture was evaluated, as other studies have described that both co-culture of MSCs and the addition of early acting cytokines have a positive effect on the ex vivo expansion of HSPCs [1,29,30]. Here, we confirmed that the recurrent addition of cytokines is beneficial for HSPCs proliferation and co-culture stability. Finally, the culture time was determined, and we found that after 7 days, the doubling rate is between 10 and 15 times, a period that has been previously employed in other studies [7,14].

Once culture conditions were defined, the functionality of HSPCs was evaluated. Upon analyzing the proliferation patterns of HSPCs cultured with early acting cytokines and co-cultured with each of the studied MSC sources, an increase in proliferation was observed in both cases (Figure 1). This result confirmed that the presence of MSCs during cell expansion promotes the proliferation of HSPCs. Furthermore, it was observed that co-cultures with MSCs from perinatal tissues achieved proliferation rates similar to those of BM-derived MSCs, confirming that these cells have a supportive capacity comparable to the standard method. Similarly, other studies have reported significant improvement in the ex vivo expansion of HSPCs using perinatal tissues as a source of MSCs [23,31]. Remarkably, MSCs from UCB have been underutilized for the ex vivo expansion of HSPCs, and according to our results, they exhibit a support capacity similar to BM-derived MSCs. This suggests a novel application of UCB-derived MSCs as a tool for HSPCs expansion. Despite displaying similarities, it was found that WJ-derived MSCs have a lower support capacity for proliferation compared to BM and UCB. This result was confirmed through proliferation studies using CFSE labeling (Figure 1). These proliferation patterns may be related to the secretory profile compared to the other sources of MSCs (Figure 5).

The in vitro migration of HSPCs towards chemotactic stimuli present in the bone marrow niche (SDF-1 and VCAM-1) allows for a preliminary assessment of their homing and engraftment capacity during HSPC transplantation, ensuring hematopoietic reconstitution [32,33]. In this study, the migratory capacity of both freshly isolated and co-cultured and expanded HSPCs was evaluated. It was found that despite statistically significant differences compared to freshly isolated HSPCs not being observed, there is a tendency toward diminished migration towards chemotactic and adhesion stimuli after the expansion process (Figure 3). This decrease could be related to proliferation and differentiation mechanisms derived from the co-culture expansion with MSCs. Such functional changes may inversely affect the response to chemotactic signals inherent to the bone marrow microenvironment. Several studies have demonstrated that stimulation with proinflammatory cytokines such as IL-6 and IL-8 on HSPC, primarily secreted by MSCs in the bone marrow niche, promotes the proliferation and differentiation of HSPCs, thereby facilitating their extravasation. These processes influence the response to chemotactic and adhesion stimuli, which are characteristic of primitive hematopoietic progenitor cells [34]. On the other hand, it has been described that the presence of high oxygen concentrations in the bone marrow niche enhances migration away from the bone marrow, progressively diminishing the primitiveness of HSPCs [35,36].

One of the main challenges during the ex vivo expansion processes of HSPCs is the activation of cellular differentiation mechanisms, leading to changes in the capacity and timing of hematopoietic regeneration once the cells have been transplanted [8,34,37]. The evaluation of the expression of markers related to the primitiveness of CD34+ HSPCs showed that the expression of this marker tends to decrease after the expansion process, with slightly higher expression in the case of co-culture with MSCs. This suggests that MSCs may partially regulate the primitiveness of HSPCs. Similar variations in CD34 expression have been reported in other studies [23], which could be related to the main drawback of this expansion technique, i.e., the expansion of differentiated cells. However, there is also evidence that the infusion of expanded hematopoietic committed progenitors (which express low levels of CD34) could reduce the rates of graft failure and the time of neutrophil engraftment, which are the principal disadvantages of UCB transplantation [38]. On the other hand, an increase in the expression of the differentiation-related marker CD38 was observed in both monoculture-expanded HSPCs and UCB-MSCs. This increase could be associated with cellular differentiation processes, while BM and WJ may partially control this process (Figure 3), inducing the differentiation of HSPCs towards more lineage-committed progenitors. Additionally, the evaluation of markers related to lymphoid lineage differentiation (CD3 and CD7) showed a small increase, suggesting a degree of commitment toward this lineage. Such a commitment toward the lymphoid lineage has been reported when HSPCs receive stimulation from proinflammatory cytokines such as IL-6 and GM-CSF [39,40]. In contrast, the myeloid lineage, indicated by the CD123 marker, did not show significant changes. Based on these results, it can be concluded that MSCs from perinatal tissues could exhibit similar behavior in terms of controlling the differentiation of BM-derived HSPCs, which correlates with earlier neutrophil engraftment in preclinical and clinical models [38]. However, tighter control of culture conditions, such as oxygen availability, is necessary, as other studies have found that maintaining ex vivo expansion techniques under hypoxic conditions largely preserves cell primitiveness and ensures HSPC proliferation [37,41,42].

To verify the functionality of HSPCs after they had been co-cultured with MSCs from different tissues, the clonogenic capacity of these cells was assessed. In this case, it was found that expanded HSPCs are capable of proliferating and forming colonies of different lineages. This suggests that the cells proliferating within the culture could be lineage-committed progenitor cells. This means that HSPCs may be exhibiting some degree of differentiation; however, it is not pronounced enough to inhibit their clonogenic capacity. Additionally, this property may be regulated by BM-MSCs and perinatal tissue MSCs.

The release of soluble factors by MSCs is the main mechanism for regulating HSPCs homeostasis, both in the bone marrow niche and in ex vivo expansion systems [43]. Proteomic and secretomic studies for characterizing these soluble factors in perinatal tissues are recent. The secretion of cytokines and soluble factors such as IL-1, IL-2, IL-6, IL-7, IL-8, IL-12, IL-15, HGF, CCL2, MIP-1, RANTES, PDGF-AA, VEGF, bFGF, and TGF-β has been reported [41,44]. All these soluble factors are related to the maintenance, proliferation, and differentiation of HSPCs in the bone marrow niche [45,46,47]. In the present study, it was demonstrated that cytokines such as EGF, FGFb, GM-CSF, G-CSF, and VEGF are secreted proportionally by the three sources of MSCs. These cytokines have been described to be involved mainly in processes such as cell homing, the regulation of apoptotic events, and inflammatory responses [48]. Therefore, within the expansion process with MSCs from perinatal tissues, these cytokines could be participating in this process. However, intriguingly, previous studies have linked EGF and FGF-b to the inhibition of HSPC proliferation [47].

On the other hand, and differentially, cytokines such as IL-6, IL-8, and CCL2 were found to be significantly secreted when HSPCs interacted with both MSC-BM and MSC-UCB. It has been described that these cytokines are secreted by MSCs in inflammatory and oxidative stress processes, inducing the proliferation and differentiation of HSPCs [49,50,51,52]. This result could be correlated with variations in the expression of differentiation markers CD38 and CD3, which are related to both myeloid and lymphoid lineage differentiation processes. Similarly, when analyzing the secretory profile of HSPCs+WJ-MSC cultures, it is evident that CCL2, EGF, and IL-8 did not show changes in their secretion levels. It is highly probable that the secretory profile of these cytokines is directly related to the support capacity of MSCs from this source, which showed lower proliferation percentages and lower expression of differentiation markers like CD38. In this regard, CCL2 has been described as having an effect on extravasal migration and cell differentiation, promoting the maintenance of migratory capacity and at the same time being involved in the differentiation of HSPCs into monocytes [53].

It is highly likely that the notable difference in the secretion of this cytokine contributes significantly to the lower performance of Wharton’s jelly MSCs that is consistently observed in some of the properties evaluated in this study. Additionally, our results indicate that the secretion levels of GM-CSF, G-CSF, and FGF from expanded cells are close to zero, and the secretion of these cytokines in co-cultures is considerably higher than in MSC cultures (Figure 5). This suggests that the secretion of these cytokines seems to be exclusive to the cellular context of co-cultures and that their individual or synergistic effect would be potent enough to have noticeable effects with relatively low secretion levels.

Finally, our clustering analysis revealed that the co-culture strategy indeed has a clear and differential effect on the secretory profile of the cells (Figure 5D). Furthermore, the clustering strategy employed fails to distinguish between bone marrow and umbilical cord blood co-cultures but does distinguish between these two sources and Wharton’s jelly (Figure 5E and Figure 6B), which aligns with the results of proliferation and differentiation assays. Remarkably, the observed secretory profile supports the proposal of a potential application of umbilical cord blood MSCs for HSPC expansion due to their similarity to bone marrow MSCs. In fact, the variables of the proliferation rate and IL-8 secretion contribute significantly to the two principal components captured by the principal component analysis, reaffirming their aforementioned importance in the cell-expansion strategy. In this regard, it is possible that expanding the scope of future studies to evaluate the secretory profile of co-cultures by including a greater number of cytokines could lead to conceptual consensus about the role of microenvironmental signals in the performance of HSPC expansion strategies and other phenomena. Finally, the first and second principal components identified in our analysis (Dim1 and Dim2) encompass only 67% of the total variability in this study (Figure 6). Therefore, approximately one-third of the observed variability throughout the study cannot be directly ascribed to any of the 17 variables encompassed in the PCA. It is plausible that in forthcoming investigations, this residual portion of overall variability could be attributed to the donors of HSPCs and MSCs.

## 4. Materials and Methods

### 4.1. Donor Selection

This research project received approval from the Ethics Committee of Subred Norte de Bogotá (Resolution: SNACEI-087). All procedures were conducted following the signing of appropriate informed consent forms. The inclusion criteria encompassed healthy pregnant individuals with uncomplicated singleton pregnancies, aged ≥18 years, with a gestation of ≥34 weeks, and no family history of hereditary cancer or genetic disorders of the lymphohematopoietic system. Additionally, individuals with no record of diseases potentially transmitted through blood, no infection or fever during labor or postpartum, and no severe anemia were included. Umbilical cord fragments were collected in 50 mL tubes containing sterile saline solution, while umbilical cord blood was gathered in Terumo or Grifols blood donation bags. Tissue collection adhered to NETCORD-FACT standards, and the transportation of tissues to the laboratory was completed within the initial 18 h.

Furthermore, authorization for bone marrow sample collection was granted by the ethics committee of the Fundación Hospital de la Misericordia (Minutes N° 45 354 20R). These samples were sourced from donors who underwent bone marrow collection for transplantation to related patients. Consent was obtained from adult donors through informed consent, and for underage donors, consent was secured from their legal guardians along with their personal assent.

### 4.2. Isolation of Cell Lines from BM MSCs

For the isolation of BM-derived MSCs, samples underwent mononuclear cell separation using a Ficoll density gradient. The upper fraction was collected and cultured at a cell density of 1 × 10^6^ cells/mL in 75 cm^2^ culture flasks containing DMEM medium supplemented with 10% FBS. A medium change occurred every 48 h until the emergence of fibroblast-like cell colonies was observed. Subsequently, the cells were expanded and subsequently cryopreserved.

### 4.3. Isolation of Cell Lines from WJ MSCs

The umbilical cord was sectioned into 10 cm fragments, and each fragment was transversely divided to eliminate blood vessels, resulting in explant sections of Wharton’s jelly. These explants were then cultured through adhesion in 10 cm^2^ Petri dishes with DMEM (Gibco^®^, Thermo Fisher, Waltham, MA, USA) culture medium supplemented with 10% FBS. The medium was changed every 48 h until the adhesion of fibroblast-like cells and the formation of colonies were noted. These cells were expanded and subsequently cryopreserved.

### 4.4. Isolation of Cell Lines from UCB MSCs

Mononuclear cells were isolated utilizing a Ficoll density gradient. These cells were then cultured at a density of 1 × 10^6^ cells/mL in DMEM culture medium supplemented with 10% FBS and dexamethasone at a concentration of 100 nM for 72 h. Following this, the culture medium was changed every 48 h until fibroblast-like colonies formed. The cells were then expanded in DMEM medium with 10% FBS and subsequently cryopreserved.

### 4.5. Immunophenotypic Characterization of MSCs

The immunophenotyping process was conducted using flow cytometry (BD FACSCanto II, BD Biosciences, San Jose, CA, USA), adhering to the established criteria outlined by the International Society for Cellular Therapy (ISCT). The quantitative analysis of positive markers encompassed CD105-allophycocyanin (APC), CD90-fluorescein isothiocyanate (FITC), CD73-APC, and CD44-phycoerythrin-cyanine 7 (PECy7), while a low expression or lack expression of markers included CD45-phycoerythrin (PE), CD54-Pacific Blue and HLA-DR-PECy7. The antibodies employed were sourced from BioLegend (San Diego, CA, USA).

### 4.6. CD34+ HSPCs Immunomagnetic Isolation

Mononuclear cells were isolated from UCB using the Ficoll density gradient method as previously described. CD34+ cells were purified through positive selection, utilizing immunomagnetic columns (MACS Miltenyi Biotec, San Jose, CA, USA). In brief, after Ficoll separation, mononuclear cells were suspended in PBS with 0.5% BSA and 2 mM EDTA. Cells were incubated with Fc antibodies for 15 min, followed by magnetic bead-conjugated CD34 antibodies for 50 min at 4 °C. The cell suspension was then passed through LS columns (Miltenyi Biotec, Bergisch Gladbach, Germany). After column washing with BSA/EDTA/PBS, the column was detached from the magnet, and the CD34+ HSPCs were collected. The percentage of CD34+ cells was determined using flow cytometry, with CD34-PerCP-Cyanine5.5 antibodies (Bio Legend, San Diego, CA, USA).

### 4.7. Irradiation of MSCs

MSCs derived from BM, UCB, and WJ (1 × 10^6^ cells) were cultured in sterile 10 cm^2^ Petri dishes until reaching 80% confluence. A single irradiation cycle was performed for each MSC source, exposing the MSCs to 25 Gy for 20–30 min using a gamma irradiation device equipped with a Cesium-137 source. This irradiation step was carried out approximately 4 h prior to initiating the co-culture.

### 4.8. Co-Culture of MSCs with HSPCs

Following the irradiation of cells and the isolation of CD34+ HSPCs from UCB, 1 × 10^5^ cells/mL of CD34+ HSPCs were seeded in RPMI 1640 medium (Gibco, USA) supplemented with 10% FBS and early acting cytokines (SCF, Flt3, and TPO (50 ng/mL)) (Gibco, USA) for a duration of 7 days. Additional supplemented medium was introduced on day 4 of the co-culture period. On day 7, cells were harvested for viability assessment, proliferation capability analysis, migration evaluation, and differentiation characterization. In certain experiments, conditioned media were collected both from MSCs in monoculture and co-culture setups, aiming to evaluate the secretome through luminex analysis.

### 4.9. Assessment of Proliferation Capacity

The proliferation capacity was determined through 2× trypan blue staining and manual counting in a Neubauer chamber. Additionally, proliferation was evaluated by counting the number of cell divisions using flow cytometry based on carboxyfluorescein diacetate succinimidyl ester (CFSE) labeling, following the manufacturer’s guidelines (Life Technologies, Carlsbad, CA, USA). HSPCs were suspended in a solution of 5 μM CFSE in PBS with 0.1% BSA and incubated for 10 min at 37 °C. Subsequently, the cells were resuspended in RPMI medium supplemented and incubated for 5 min at 4 °C, followed by washing with PBS 1× and centrifugation at 400× *g* for 7 min. The cells were then suspended in RPMI medium with 10% FBS and incubated for 20 min at 37 °C. Co-culture with MSCs was then initiated. To establish the zero-proliferation point, HSPCs were cultured under starvation conditions for 24 h. At the endpoint of proliferation assessment, cells were cultured without CFSE labeling.

### 4.10. Evaluation of Migration Capacity

Migration capacity was evaluated using Transwell chambers (Corning Costar, Tewksbury, MA, USA) with a diameter of 6.5 mm and a pore size of 5 μm. The inserts were pre-incubated for one hour in a migration solution (RPMI 1640/2% BSA). CD34+ HSPCs (1 × 10^5^) obtained from different experimental conditions were cultured and introduced into the upper chamber. In the lower chamber, 600 μL of migration buffer supplemented with VCAM-1 was added, and a chemotactic stimulus was applied using a migration buffer supplemented with SDF-1. Subsequently, the plates were incubated for 3 h at 37 °C and 5% CO_2_. The cells that migrated to the lower chamber were collected and quantified using flow cytometry at predefined time intervals and flow rates. The migration percentage was calculated by dividing the number of migrated cells by the total number of cells and multiplying by 100.

### 4.11. Determination of Differentiation Capacity

The expression of surface markers related to stemness (CD34-PercP-Cy5.5 and CD38-APC) and lineage-committed progenitors (CD123-PE, CD7-FITC, CD3-BV421) was assessed using flow cytometry. The positioning of gates was established using unmarked cells as the negative control. All antibodies were sourced from Biolegend (San Diego, CA, USA).

### 4.12. Determination of Clonogenic Potential

The in vitro clonogenic capacity of HSPC was determined based on the number of colonies generated from equal numbers of cells seeded per milliliter (150 cells/mL). Fresh isolated and purified cells were seeded in Iscove’s Modified Dulbecco’s Medium (IMDM; Gibco, USA) supplemented with SFB 2% and methylcellulose medium without erythropoietin (Methocult H4035 Optimum Without EPO; Stem Cell Technologies, Vancouver, BC, Canada). The plates were incubated in a humidified 37 °C incubator for 14 days. For isolated cells co-cultured with MSCs, the same procedure was followed with the initiation of co-culture after 7 days of initial culture.

### 4.13. Identification and Quantification of Cytokines in MSC Monoculture and Co-Culture Conditioned Media

Conditioned media from irradiated MSCs were collected 24 h post-irradiation and cryopreserved at −20 °C. For co-culture samples, the medium was collected after 7 days of culture and centrifuged at 500× *g* for 7 min to separate the CD34+ HSPCs, and the resulting supernatant was cryopreserved at −20 °C until use. All media underwent filtration through a 0.22 μm filter (Millipore, Burlington, MA, USA). Duplicate samples from the three evaluated treatments (HSPC+ cytokines, MSC, and HSPC+ MSC) sourced from the three studied origins (BM, UCB, WJ) were subjected to analysis using the multiparametric suspension luminometry assay (Luminex) with the Luminex^®^ Performance Human XL Cytokine Premixed Kit, following the manufacturer’s instructions. The panel comprised 8 cytokines: IL-6, IL-8, CCL2, basic FGF, VEGF, GM-CSF, and G-CSF. In brief, the reagents provided in the kit were prepared as per the instructions. On the day of the assay, the samples were centrifuged at 16,000× *g* for 4 min and immediately diluted at a 1:2 ratio with the kit’s diluent. Subsequently, 50 µL per well was added to a 96-well plate, also provided in the kit. Then, 50 μL of the cytokine cocktail was added to each well, and the plate was incubated for 2 h at room temperature, shielded from light, under constant agitation. After incubation, the plate was washed three times with a wash solution, and 50 µL of biotinylated antibody was added to each well. The plate was incubated for 1 h at room temperature in the dark and washed twice with the wash solution. Following this, 50 µL of streptavidin-PE was added to each well, and the plate was incubated for 30 min at room temperature in darkness. Subsequently, the wells were washed three times and filled with 100 µL of wash solution, and the plate was incubated for 2 min at room temperature in darkness. Finally, the plate was read with the Luminex^®^ 100/200™, and the obtained data were analyzed using a 5-parameter logistic calibration curve. It is important to note that the blanks, controls, and standards included in the kit were analyzed following the same procedure described above without any discernible performance issues.

### 4.14. Variable Correlation Analysis

For the variable correlation analysis, the data were normalized to ensure comparability among variables. Subsequently, the data derived from the Luminex analysis were grouped and visualized using the pheatmap package (pheatmap, RRID:SCR_016418) [54]. Principal component analysis involved compiling data from all presented assays into a matrix, focusing on data from the HSPC donors who underwent all functional assays. This analysis was conducted using the factoextra package (factoextra, RRID:SCR_016692) [55], which assessed the contribution of each principal component to the total variability and visually represented these components to discern patterns and relationships among variables within the context of principal components. Both tools are implemented in R (R Project for Statistical Computing, RRID:SCR_001905) [56].

### 4.15. Data Analysis

A normality analysis of the data was conducted using the Kolmogorov–Smirnov test. The Kruskal–Wallis, Wilcoxon, and Mann–Whitney tests were employed to evaluate the statistical significance of the observed differences. To determine the correlation between variables, a linear model was constructed, and the statistical significance was verified using Pearson analysis. Finally, it is important to note that the logistical constraints, the time required to obtain confluent MSCs from the three studied sources and the collection of USCU for fresh CPH isolation limited the execution of a greater number of experiments for the described assays.

## 5. Conclusions

The results of this study demonstrate that perinatal tissues, which are commonly considered as waste products after childbirth, can be a suitable source of MSCs. These cells provide support for the ex vivo expansion of fresh CB-derived HSPC cells, offering an alternative source of MSCs, in addition to BM, which has already been extensively studied. These techniques for collection and isolation are less invasive and readily available, making them a favorable option for patients requiring transplants.

For future studies in this field focusing on expansion techniques using MSCs from perinatal tissues, we suggest considering the tight control of co-culture system conditions, such as irradiation of MSCs and co-culture under hypoxic conditions, to better manage oxidative stress levels and more accurately simulate the bone marrow niche. Similarly, exploring xeno-free culture conditions is recommended. These modifications will enable better control over cell primitiveness, ensuring improved long-term cellular reconstitution outcomes in HSPC transplantation. Thus, it is suggested that future analyses incorporate a donor classification method to enhance sample and result characterization. This measure will contribute to a more precise and robust interpretation of the data.

Finally, MSCs from UCB are proposed as an alternative and underexplored source of support for HSPC expansion. Future research should delve deeper into the unique characteristics and potential of UCB-derived MSCs, including their secretory profile and impact on HSPC functionality, to fully harness their capacity for ex vivo expansion and advance their clinical application.

## Figures and Tables

**Figure 1 ijms-24-15544-f001:**
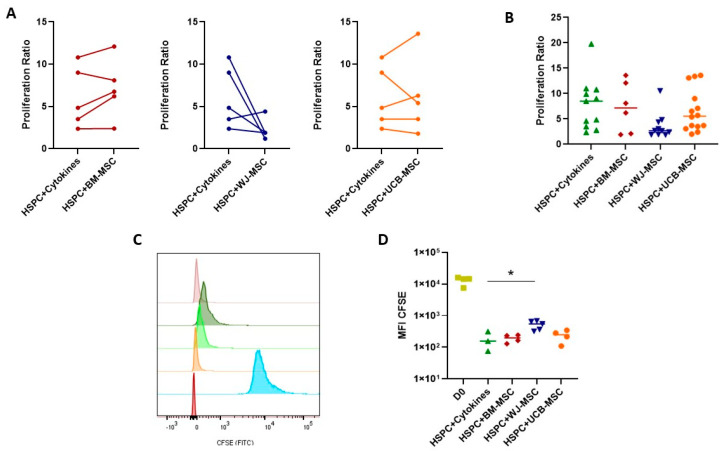
Proliferation capacity of umbilical cord blood hematopoietic stem and progenitor cells (HSPCs) co-cultured with mesenchymal stromal cells (7-day culture). A comparison is made with HSPCs in monoculture (HSPC+Cytokines, depicted in green) treated with early acting cytokines (SCF, TPO, FLT3L). Co-culture with bone-marrow-derived MSCs (HSPC+BM-MSC, depicted in red). Co-culture with Wharton’s-jelly-derived MSCs (HSPC+WJ-MSC, depicted in blue). Co-culture with umbilical-cord-blood-derived MSCs (HSPC+UCB-MSC, depicted in orange). (**A**) Paired analysis of donors in mono- and co-culture. (**B**) Proliferation analysis of different donors under mono- and co-culture conditions. (**C**) Proliferation analysis of CFSE-labeled HSPCs and Mean Fluorescence Intensity (MFI) analysis via flow cytometry. Red: Unlabeled cells. Blue: Cells synchronized via 24 h starvation. Orange: HSPC+Cytokines. Light Green: HSPC+BM-MSC. Dark Green: HSPC+WJ-MSC. Violet: HSPC+UCB-MSC (Representative image). (**D**) Analysis of the MFI of CFSE-labeled HSPCs under different culture conditions. Statistical analysis performed using the Wilcoxon paired test (non-significant) and Mann–Whitney test (*p* < 0.05). * Data statistically significant. Values are presented as the number of times HSPCs are doubled.

**Figure 2 ijms-24-15544-f002:**
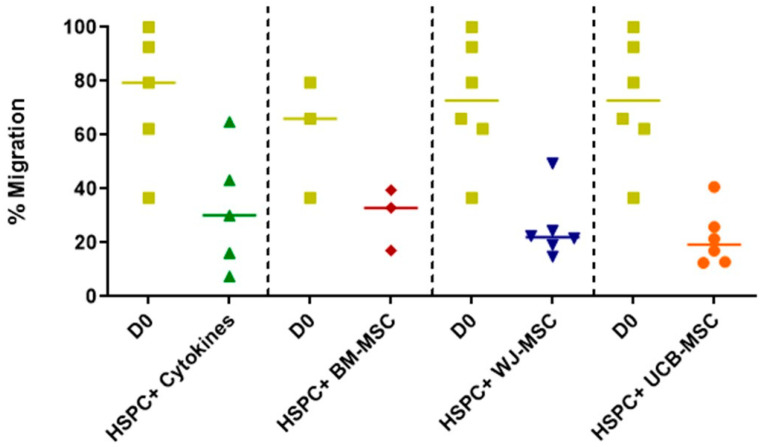
Migration capacity of umbilical cord blood hematopoietic stem progenitor cells (HSPCs) in co-culture with mesenchymal stromal cells (7-day culture). The migratory behavior of HSPCs co-cultured with mesenchymal stromal cells (MSCs) is assessed using a transwell system and chemotactic/adhesive stimuli for a duration of three hours, followed by flow-cytometry-based cell counting. The migration capacity of the co-cultured HSPCs is compared with that of HSPCs cultured alone with early acting cytokines (HSPC+Cytokines, depicted in green). Additionally, co-cultures are performed with bone-marrow-derived MSCs (HSPC+BM-MSC, shown in red), Wharton’s-jelly-derived MSCs (HSPC+WJ-MSC, depicted in blue), and umbilical-cord-blood-derived MSCs (HSPC+UCB-MSC, represented in orange). Statistical analysis is conducted using the Kruskal–Wallis test on unpaired data, and no significant differences in migration capacity are observed among the different co-culture conditions.

**Figure 3 ijms-24-15544-f003:**
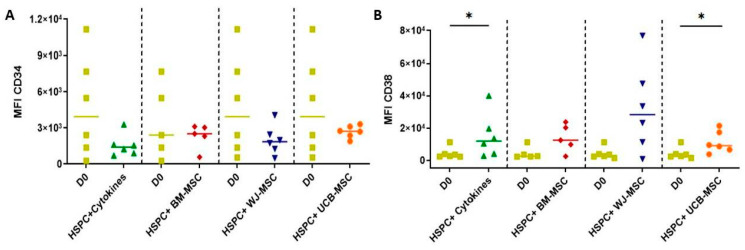
Evaluation of the expression of primitive markers in umbilical cord blood hematopoietic stem progenitor cells (HSPCs) in co-culture with mesenchymal stromal cells (MSC) (7-day culture). The flow-cytometry-based assessment of primitive and differentiation markers in HSPCs cultivated with MSCs is depicted. Expression levels of these markers in freshly isolated HSPCs are quantified and juxtaposed with HSPCs in monoculture (HSPC+Cytokines, green) treated with early acting cytokines (SCF, TPO, FLT3). Co-culture with bone-marrow-derived MSCs (HSPC+BM-MSC, red). Co-culture with Wharton’s-jelly-derived MSCs (HSPC+WJ-MSC, blue). Co-culture with umbilical-cord-blood-derived MSCs (HSPC+UCB-MSC, Orange). (**A**) Quantification of Mean Fluorescence Intensity (MFI) for CD34. (**B**) Median Fluorescence Intensity (MFI) for CD38. Statistical analysis was performed using the Wilcoxon test for paired data (*p* < 0.05). * Data statistically significant.

**Figure 4 ijms-24-15544-f004:**
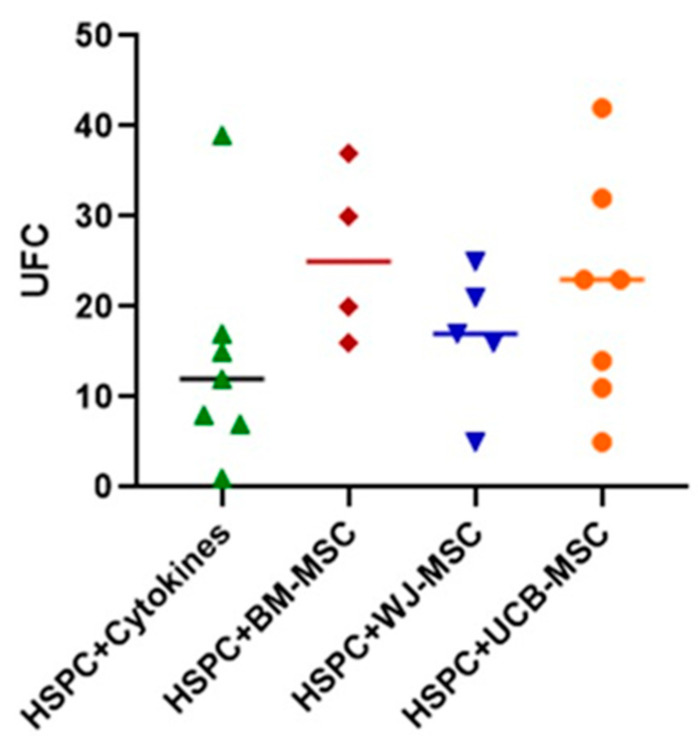
Evaluation of the clonogenic potential of umbilical cord blood hematopoietic stem progenitor cells (HSPC) in co-culture with mesenchymal stromal cells (MSC) (14-day culture). The clonogenic potential of CD34+ cells expanded in co-cultured with mesenchymal stromal cells (MSCs) is assessed by counting Colony-Forming Units (CFUs) formed in methylcellulose culture from HSPCs expanded in co-cultured with MSCs. The clonogenic capacity of the co-cultured HSPCs is compared with that of HSPCs cultured alone with early acting cytokines (HSPC+Cytokines, depicted in green). Additionally, co-cultures are performed with bone-marrow-derived MSCs (HSPC+BM-MSC, shown in red), Wharton’s-jelly-derived MSCs (HSPC+WJ-MSC, depicted in blue), and umbilical-cord-blood-derived MSCs (HSPC+UCB-MSC, represented in orange). Statistical analysis was conducted using the Kruskal–Wallis test for unpaired data (*p* < 0.05).

**Figure 5 ijms-24-15544-f005:**
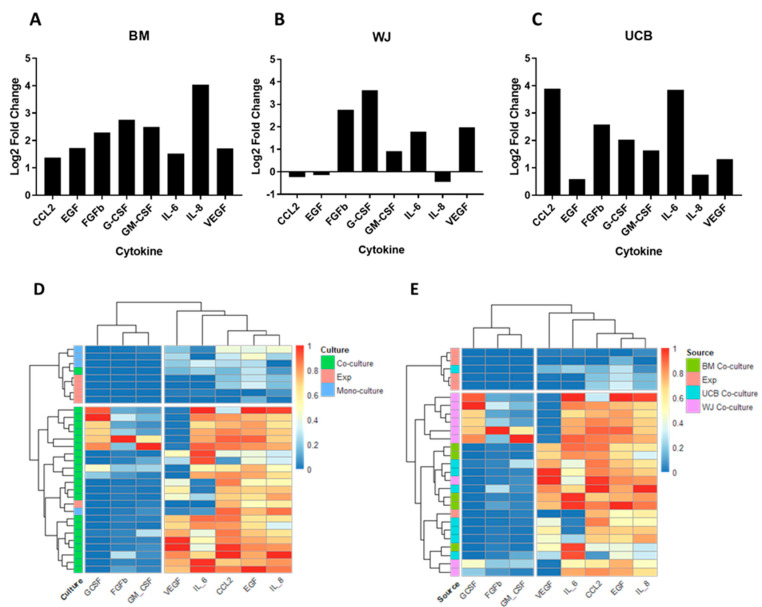
Cytokine secretion profile of HSPCs + MSCs co-cultures. The change in secretion levels of eight cytokines is presented as the Log2 of the fold change between hematopoietic progenitor cell and mesenchymal stromal cell co-cultures from each tissue. (**A**) Bone marrow. (**B**) Wharton’s jelly. (**C**) Umbilical cord blood. Samples are grouped according to their cytokine expression profile. Each row represents a sample, and each column represents a cytokine. Both samples and cytokines were grouped based on their similarity. (**D**) Sample grouping by treatment. (**E**) Sample grouping by source. Exp: HSPC+Cytokines w/o MSC culture.

**Figure 6 ijms-24-15544-f006:**
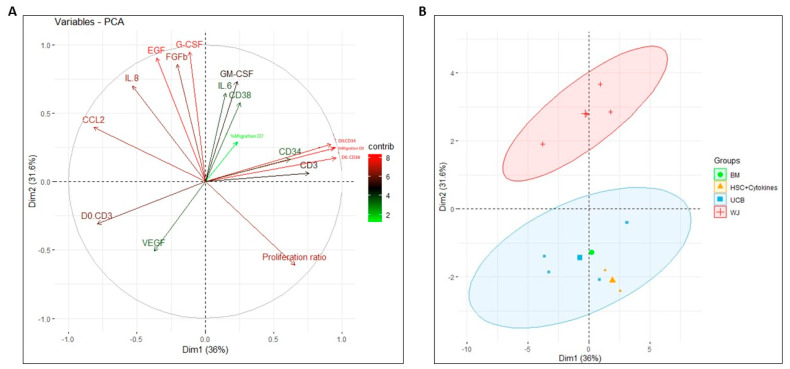
Principal component analysis (PCA) according to HSPCs properties. A two-dimensional PCA was conducted with donors of HSPCs who underwent comprehensive functionality assays (proliferation, migration, differentiation, and secretory profile). (**A**) Correlation circle of evaluated variables according to contribution scale, where variables with a higher contribution are shown in red and those with lower contribution are shown in green. (**B**) Two-dimensional PCA based on MSC expansion strategy (co-cultures with MSCs from different sources versus HSPCs expanded with early action cytokines), displaying the contribution of these variables to the principal components and allowing the clear grouping and discrimination of Wharton’s jelly data (red) from other groups (blue).

## Data Availability

Raw data for the article are available upon request to the corresponding authors.

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
