# Peer review of "Mesenchymal Stromal Cells from Perinatal Tissues as an Alternative for Ex Vivo Expansion of Hematopoietic Progenitor and Stem Cells from Umbilical Cord Blood"

_ijms, 2023, doi:10.3390/ijms242115544_

Round 1
Reviewer 1 Report
In the manuscript Bonila and co-workers claim that perinatal mesenchymal stem cells (MSC) are an efficient support for the expansion of the hematopoietic stem cells. Notably , CBU-derived MSCs show asimilar supportive capacity to BM-derived MSC for the HSC maintenance and expansion. However there are several conceptual and writing issues needed to be corrected in order that this manuscript could be considered for publication.
1. CBU supportive role–of mesenchymal stromal cell for the hematopoietic stem and progenitors cells is already well documented ( DOI: 10.1089/ten.tea.2013.0073 ; DOI: 10.1186/s13287-021-02474-8 DOI: 10.1186/s13287-015-0194-y) . Thus, the authors should highlight much more clearly the novelty that their manuscript bring in the field.
2. There is a serial conceptual issue about HSC definition used in the study. Except in the title, all over the manuscript the authors name the CD34+ hematopoietic cells “HSC” which is well known to the be completelymistaken. First, the CD34+ cells include mostly the hematopoietic progenitors with very rare HSC. For the cord blood it was determined that HSC are present with no more than…1-2% of the Lin-Cd34+Cd38- population (Yahata et al, 2003; Ivanovic&Boiron, 2009)
3.Tthe author talk about the expansion of the HSC while it was also well known that the big problem with the expansion protocols is that, along the expansion, hematopoietic cells differentiate in more matures cells. That’s why in order to estimate the presence of the HSC in the cell populationaccurate tests are needed otherwisewe cannot conclude that we got the HSC. The golden standard is SRC activity or vitro test, that are less reliable based on re-plating clonogenic activity and/ or phenotypic characterization; keeping in mind that phenotypic characterization w/o associated functional tests are only approximations; of course, phenotypic markers that are identified to allow selection of the population higlly enriched in HSC are documented. (Notta et al, 2011; Takahashi et al, 2014; Majeti et al,2007). Estimation is based only on the CD38 and CD34 marker expression that are present in the hematopoietic population compose the majority of the progenitors cells that are far from being accurate markers proving the presence of the HSC . So far the only population almost entirely composed of HSCs is Lin−CD34+CD38−CD45RA−Thy+RholoCD49f+ cells [Notta et al, 2011).
4. Based on the self-renewal and multipotency as the hallmark and sine qua non charasteristic of the stem cells , mesenchymal stem cells represent a minority in the heterogeneous population of mesenchymal stromal cells readily isolated from many different adult tissues including the perinatal one . Thus, it is not suitable to use the term mesenchymal stem cell for deciphering the whole mesenchymal stromal cells composed of different types of cells, including committed progenitor cells and post-progenitor cells [3]. (Loncaric et al, 2018; Muraglia et al, 2000; Hammoud, M et al J Cell Physiol. 2012, 227, 2750; (; (5) Mylotte, LA et al Stem Cells 2008, 26.
Additional coments:
-Authors wrote about ISCT criteria for identification of MstroC that comprise behind the plastic adherence and marker expressions the estimation of the multipotent capacity which was not done in the manuscript.
-Line 78/79; Alternate strategies in-78 involve expanding HSCs using cytokines, copper chelation, or co-culturing with mesenchy-79 mal stem cells (MSCs) [1,14]. The reference Hammoud et al, 2012 should be added.
-All over the text, there is a confusion about the figure’s denomination: it is not clear which are the supplementary which are the main figures; Also, there are no figure legends for the supplementary figures.
-The results have to be presented in a way that is much more accurate. For instance; In
Determination of the Expression of Primitive and Lineage-Committed Markers in Hematopoi-243 etic Stem Cells Cultured with Mesenchymal Stromal Cells from Perinatal Tissues
do sports represent individual experiments? If it is the case, for the expression of CD19, in figures3 there is only 1 exp presented?!
In the manuscript Bonilla and co-workers claim that perinatal mesenchymal stem cells (MSC) are an efficient support for the expansion of the hematopoietic stem cells. Notably , CBU-derived MSCs show asimilar supportive capacity to BM-derived MSC for the HSC maintenance and expansion. However there are several conceptual and writing issues needed to be corrected in order that this manuscript could be considered for publication.
1. CBU supportive role–of mesenchymal stromal cell for the hematopoietic stem and progenitors cells is already well documented ( DOI: 10.1089/ten.tea.2013.0073 ; DOI: 10.1186/s13287-021-02474-8 DOI: 10.1186/s13287-015-0194-y) . Thus, the authors should highlight much more clearly the novelty that their manuscript bring in the field.
2. There is a serial conceptual issue about HSC definition used in the study. Except in the title, all over the manuscript the authors name the CD34+ hematopoietic cells “HSC” which is well known to the be completelymistaken. First, the CD34+ cells include mostly the hematopoietic progenitors with very rare HSC. For the cord blood it was determined that HSC are present with no more than…1-2% of the Lin-Cd34+Cd38- population (Yahata et al, 2003; Ivanovic&Boiron, 2009)
3.Tthe author talk about the expansion of the HSC while it was also well known that the big problem with the expansion protocols is that, along the expansion, hematopoietic cells differentiate in more matures cells. That’s why in order to estimate the presence of the HSC in the cell populationaccurate tests are needed otherwisewe cannot conclude that we got the HSC. The golden standard is SRC activity or vitro test, that are less reliable based on re-plating clonogenic activity and/ or phenotypic characterization; keeping in mind that phenotypic characterization w/o associated functional tests are only approximations; of course, phenotypic markers that are identified to allow selection of the population higlly enriched in HSC are documented. (Notta et al, 2011; Takahashi et al, 2014; Majeti et al,2007). Estimation is based only on the CD38 and CD34 marker expression that are present in the hematopoietic population compose the majority of the progenitors cells that are far from being accurate markers proving the presence of the HSC . So far the only population almost entirely composed of HSCs is Lin−CD34+CD38−CD45RA−Thy+RholoCD49f+ cells [Notta et al, 2011).
4. Based on the self-renewal and multipotency as the hallmark and sine qua non charasteristic of the stem cells , mesenchymal stem cells represent a minority in the heterogeneous population of mesenchymal stromal cells readily isolated from many different adult tissues including the perinatal one . Thus, it is not suitable to use the term mesenchymal stem cell for deciphering the whole mesenchymal stromal cells composed of different types of cells, including committed progenitor cells and post-progenitor cells [3]. (Loncaric et al, 2018; Muraglia et al, 2000; Hammoud, M et al J Cell Physiol. 2012, 227, 2750; (; (5) Mylotte, LA et al Stem Cells 2008, 26.
Additional coments:
-Authors wrote about ISCT criteria for identification of MstroC that comprise behind the plastic adherence and marker expressions the estimation of the multipotent capacity which was not done in the manuscript.
-Line 78/79; Alternate strategies in-78 involve expanding HSCs using cytokines, copper chelation, or co-culturing with mesenchy-79 mal stem cells (MSCs) [1,14]. The reference Hammoud et al, 2012 should be added.
-All over the text, there is a confusion about the figure’s denomination: it is not clear which are the supplementary which are the main figures; Also, there are no figure legends for the supplementary figures.
-The results have to be presented in a way that is much more accurate. For instance; In
Determination of the Expression of Primitive and Lineage-Committed Markers in Hematopoi-243 etic Stem Cells Cultured with Mesenchymal Stromal Cells from Perinatal Tissues
do sports represent individual experiments? If it is the case, for the expression of CD19, in figures3 there is only 1 exp presented?!
Reviewer 2 Report
In this manuscript written by Bonilla et al, the main subject is to describe the hematopoietic support capacity of MSCs from Wharton's jelly and umbilical cord blood, evaluating the proliferation, migration and cytokine secretion capacity of CD34+ cells from umbilical cord blood. Although the article is well written and has interesting results, it is necessary to determine the presence of hematopoietic stem and progenitor cells in the cocultures analyzed. The following comments should be taken into consideration.
1. In Supplementary Figure 1, HLA-DR expression is high in MSC-BM under basal conditions, i.e. in the absence of stimuli such as induction with INF-γ. This result is striking given that low expression of this histocompatibility molecule has been reported for this type of MSCs, in fact, this low expression is related with an important characteristic of these cells which is low immunogenicity. It is necessary to clarify in this regard.
2. As indicated by the authors, in addition to the immunophenotype analysis, it is necessary to determine the adipogenic, osteogenic and chondrogenic differentiation potential of the MSCs obtained from the different sources, in accordance with the criteria for characterizing MSCs established by the ISCT. It is necessary to perform the corresponding experiments.
3. The CD34+ hematopoietic cell population includes two compartments within the hematopoietic system, such as hematopoietic stem cells and hematopoietic progenitor cells, because of this it is not possible indicate that CD34+ cells are a stem cell population. It is necessary to consider this aspect throughout the text, including the subheadings in the results section.
4. Correct in Figures 2 and 3 the subtitles of the graphs corresponding to HSC+WJ-MSC and HSC+BM-MSC, which are interchanged as described in the figure caption.
5. The authors point out that the presence of MSCs during cell expansion promotes the proliferation of hematopoietic stem cells (page 11, lines 431, 432), however, to determine that the number of HSCs is increased, it is necessary to evaluate how many of the cells that proliferate in the cultures are really of stem characteristics and for this purpose it would be necessary to obtain the cellular index of hematopoietic reconstitution in SCID mice.
6. Given that the authors obtained hematopoietic populations with markers of differentiated cells in their co-cultures, this indicates that there is no expansion of CD34+ hematopoietic cells and therefore the observed increase in cell proliferation is constituted by differentiated hematopoietic cells, which suggests that this strategy of expanding primitive hematopoietic cell populations using MSCs is not feasible. This contrasts with the conclusions described in the discussion section.
7. To give significance to the study it is necessary to use strategies to detect expansion of hematopoietic progenitors (colony assay) and hematopoietic stem cells (quantification of SCID repopulating cells); it is not possible to suggest ex vivo expansion strategies for clinical application if such populations are not determined more accurately.
Reviewer 3 Report
Very insightful paper with a thorough demonstration (well controlled data) of ex-vivo HSC expansion using co-culture with perinatal MSCs.
1) Do the authors think that MSC-secretome and MSC-derived exosomes might have the same effect? This might be a less cumbersome and process-heavy way.
2) While the authors evaluated "stemness" and presence of lineage-committed progenitors, were there any ex-vivo differentiation experiments performed to demonstrate functional cells deriving from the progenitors?
Round 2
Reviewer 2 Report
No comments